# Studying item-effect variables and their correlation patterns with multi-construct multi-state models

**Tina H. Erhardt**📷, **Timo Gnambs**📷, **Marie-Ann Sengewald**📷 *

Leibniz Institute for Educational Trajectories (LIfBi), Bamberg, Germany

* marie.sengewald@lifbi.de

## Abstract

Method effects on the item level can be modeled as latent difference variables in longitudinal data. These item-effect variables represent interindividual differences associated with responses to a specific item when assessing a common construct with multi-item scales. In latent variable analyses, their inclusion substantially improves model fits in comparison to classical unidimensional measurement models. More importantly, covariations between different item-effect variables and with other constructs can provide valuable insights, for example, into the structure of the studied instrument or the response process. Therefore, we introduce a multi-construct multi-state model with item-effect variables for systematic investigations of these correlation patterns within and between constructs. The implementation of this model is demonstrated using a sample of $N = 2,529$ Dutch respondents that provided measures of life satisfaction and positive affect at five measurement occasions. Our results confirm non-negligible item effects in two ostensibly unidimensional scales, indicating the importance of modeling interindividual differences on the item level. The correlation pattern between constructs indicated rather specific effects for individual items and no common causes, but the correlations within a construct align with the item content and support a substantive meaning. These analyses exemplify how multi-construct multi-state models allow the systematic examination of item effects to improve substantive and psychometric research.

**Data Availability Statement:** This study used data of the Dutch Longitudinal Internet Studies for the Social Sciences (LISS) panel administered by Centerdata (Tilburg University, The Netherlands). The data are available freely to qualified

## Studying item-effect variables and their correlation patterns with multi-construct multi-state models

Do psychological measurements provide accurate results? The answer to this question is crucial for the meaningfulness of empirical results in psychological research and depends on the definition of latent constructs and their measurement. Already Campbell and Fiske [1] emphasized that the measurement of a latent construct can be improved by using different methods because the way a construct is measured can influence the measurement results. Accordingly, various multitrait-multimethod (MTMM) models have been developed to investigate method effects, for example, concerning different raters, tests, or item bundles with different

researchers, and the authors had no special access or request privileges. Because the data contain sensitive information, users must sign a statement to download the data from the LISS website (https://www.dataarchive.lissdata.nl/).

**Funding:** The authors received no specific funding for this work.

**Competing interests:** The authors have declared that no competing interests exist.

formulations [e.g., 2]. With multiple measurement occasions in longitudinal data, method effects can also be identified for individual items [e.g., 3,4]. Although the item-level is arguably the most detailed, so far, it has received sparse attention.

An advantage of specifying method effects across multiple items is that the systematic variation can be attributed to the pre-specified differences between the items (e.g., different raters or item formulations). This idea of traditional MTMM models can also be implemented in longitudinal data [see e.g., 5,6]. However, the assumption of a common method effect for multiple items does not hold in all empirical settings. More detailed investigations on differences between items are possible with item-specific traits [e.g., 7–10] and item-specific method effects [e.g., 11–14]. Item-specific traits disentangle situation-specific effects by modeling a latent trait for each item across different time points. Instead, method effects separate the item-specificity from effects that are common for all items and are especially suited for further investigating the item-specific effects themselves. Such investigations can help to better understand the causes for the systematic variation in each item, that are not pre-specified. Explanations for item effects can support psychometric scale construction, methodological research on the occurrence of method effects, but also substantive analysis, for instance, on the predictive validity of item effects [see e.g., 15].

Item-specific method effects have been empirically studied by Cogo-Moreira et al. [11], Geiser et al. [12], Holtmann et al. [13], and Thielemann et al. [14]. Despite adopting different approaches to model item effects (e.g., as regression residuals or as latent differences), these studies concordantly showed that modeling psychological constructs with item effects substantially improve model fit and precision. Regression residuals are well suited for investigating additive variance components, because the residual item effects and latent states are typically modeled as uncorrelated components. Yet, this assumption is not necessary and, without restrictions on the correlation structure, the residual approach is closely aligned with modeling latent differences [see e.g., 4,16]. We now draw on the potential of investigating the correlation structure in latent differences models, for gaining a better understanding on possible explanations for item effects—not only in single-construct, but also in multi-construct contexts.

Specifically, we use a novel multi-construct multi-state model with item-effect variables and examine the correlations between item-effect variables of different items, as well as between item-effect variables and latent states, both, within and between constructs. In our application, we systematically study item-effect variables in two well-being instruments measuring life satisfaction and positive affect. Next to the existence of non-neglectable item-effect variables, we investigate the specific correlation patterns to identify potential common causes of item-effect variables and discuss possible explanations for the occurrence of item effects in the two well-being instruments.

## Modeling item effects in longitudinal data

The identification of method effects on the item level requires longitudinal measurements of multi-item scales. In this context, the revised latent state-trait theory, LST-R [17 with previous developments by 18–20], can be adopted to model multiple latent state variables and item-effect variables. Thereby LST-R provides a powerful theoretical framework for defining latent variables and measurement error variables that have a clear and unambiguous interpretation [see e.g., 4].

### Multi-state model

We consider $m$ manifest variables $Y_{it}$, with $i = 1,\ldots,m$, which are assessed at $n$ different measurement occasions $t = 1,\ldots,n$. Following LST-R, each manifest variable $Y_{it}$ is decomposed

into its latent state variable $\tau_{it}$ and its measurement error variable $\varepsilon_{it}$,

$$Y_{it} = \tau_{it} + \varepsilon_{it}. \tag{1}$$

All variables can take on different values for different persons. The latent state variable $\tau_{it}$ is the conditional expectation of the manifest variable $Y_{it}$ given $U_t$, the person variable at time $t$, and $S_t$, the situation variable at time $t$,

$$\tau_{it} := E(Y_{it}|U_t, S_t). \tag{2}$$

The measurement error variable $\varepsilon_{it}$ is the difference between the manifest variable $Y_{it}$ and its latent state variable $\tau_{it}$,

$$\varepsilon_{it} := Y_{it} - \tau_{it}. \tag{3}$$

For identifying the scale of a common latent state variable $\eta_t$ for different manifest variables at time $t$, we consider a specific reference, for instance, the first manifest variable, such that,

$$\eta_t = \tau_{1t}. \tag{4}$$

Based on the theoretic variables, a multi-state measurement model can be defined. This first requires an equivalence assumption, like the assumption of $\eta_t$-congenericity, specifying each item-specific latent state variable $\tau_{it}$ as a linear function of the time point specific common latent state $\eta_t : \tau_{it} = v_{it} + \lambda_{it} \cdot \eta_t, \forall i = 1, \ldots, m, \forall t = 1, \ldots, n$. For assuring an equal measurement model for the common latent state variable at different measurement occasions, measurement invariance for all measurement occasions $t$ is required. For strong measurement invariance, the equivalence assumption for each $\tau_{it}$ is specified with the same intercept and the same factor loading at each measurement occasion, such that the time index $t$ can be omitted from these parameters,

$$\tau_{it} = v_i + \lambda_i \cdot \eta_t, \forall i = 1, \ldots, m, \forall t = 1, \ldots, n. \tag{5}$$

Furthermore, the multi-state model requires that all measurement error variables of different items and time points are uncorrelated. Latent state variables and measurement error variables must be uncorrelated at all time points, too. Some of the correlations are zero by definition, but for others model assumptions are required. These are

$$cov(\varepsilon_{it}, \varepsilon_{jt}) = 0, \forall i \neq j \text{ with } i, j = 1, \ldots, m, \forall t = 1, \ldots, n, \tag{6}$$

within a time point $t$ all measurement error variables are uncorrelated, and

$$cov(\varepsilon_{it}, \tau_{is}) = 0, \forall i = 1, \ldots, m \text{ at } s > t, \tag{7}$$

measurement error variables are uncorrelated with future states. All other correlations (i.e., between measurement error variables of different time points, as well as between measurement error variables and states of the same or earlier time points) are zero by definition as derived in the revised LST theory [17].

As implied by the model assumptions, the responses of a person that are represented by the $i = 1, \ldots, m$ manifest variables $Y_{it}$, differ only due to random measurement error and item parameters that are constant across all persons. Thus, all systematic variation in the item responses is represented in the common latent states $\eta_t$, which describe an attribute of the persons in a specific situation at time $t$ that is measured on an identical scale across all measurement occasions (i.e., on the scale of the reference item).

## Extension with item-effect variables

The assumptions of a common state variable for different manifest variables and uncorrelated measurement errors may not hold in all applications, as additional systematic variation in the observed scores can be present due to method effects. Different strategies to account for method effects in longitudinal data have been proposed [see e.g., 4 for an overview]. In line with LST-R theory, method effects can be defined as regression residuals [see e.g., 21] or as latent differences [see e.g., 16]. Both approaches do not alter the meaning of the formerly well-defined state variables, while extending the multi-state model by the inclusion of item effects [see e.g., 4]. Here, in line with Pohl et al. [16], we define method effects as latent difference variables. We consider the most fine-grained level for method effects, which is the level of individual items (respectively individual manifest variables, if these do not represent items but combined scores like item parcels). As such, we build on the work of Thielemann et al. [14], who define latent item-effect variables for dichotomous items in a probit multi-state model. We use a similar definition for latent item-effect variables $\delta_{it}$ but for continuous manifest variables. A latent item-effect variable $\delta_{it}$ for a manifest variable $Y_{it}$ is the difference between the respective latent state variable $\tau_{it}$ and the common latent state variabl $\eta_t$,

$$\delta_{it} := \tau_{it} - \eta_t. \tag{8}$$

We defined $\eta_t$ as the latent state variable of the reference item, thus, the number of item-effect variables is one less than the number of items. With this definition of item-effect variables, the latent state variable $\tau_{it}$ of each item can be decomposed into the common latent state variable $\eta_t$ and the item-effect variable $\delta_{it}$ of the item $i$:

$$\tau_{it} = \eta_t + \delta_{it}, \forall i = 2, \ldots, m. \tag{9}$$

One item-effect variable is zero for defining a reference item, for instance, of the first item $Y_{1t}$. In addition, all factor loadings are set to 1 and all intercepts to 0. As such, item-effect variables describe all systematic differences between the items and represent individual differences. This specification just separates different sources of systematic variation in item-responses without any assumptions, as it can be seen when inserting Eq 8 in Eq 9,
$\tau_{it} = \eta_t + \delta_{it} = \eta_t + \tau_{it} - \eta_t = \tau_{it}$.

The identification of item-effect variables $\delta_{it}$ requires at least three repeated measurements of the same manifest variable and the assumption of identical item-effect variables for each manifest variable across the measurement occasions, that is,

$$\delta_{it} = \delta_{is} \equiv \delta_i, \forall i = 1, \ldots, m, \forall s, t = 1, \ldots, n. \tag{10}$$

This assumption of identic item-effect variables across measurement occasions is equivalent to measurement invariance at the individual level. Meaning that the differences between individual (person-specific) item effects are stable over time ($\delta_i$ is constant within each person).

In this model, the common latent states $\eta_t$ represent an attribute of the persons in a specific situation at time $t$ that is measured on an identical scale across all measurement occasions (i.e., on the scale of the reference item); but while accounting for item-specific method effects. Systematic variation in the item responses is represented in the common latent states $\eta_t$ and the item effects $\delta_i$. Thereby, modeling item effects as stable latent differences allows for estimating the expected values and variances of the item-effect variables across all persons under investigation, as well as all correlations among latent variables.

## Investigating item effects

Our model defines item-effect variables as stable person characteristics (i.e., item-effect variables $\delta_i$ represent person-specific differences between items that are constant within each person over time). Thus, instead of viewing item-effect variables as unwanted sources of variation, they can be investigated for a better understanding of the focal construct and the response process in multi-item scales. We expand previous applications by systematically investigating item-effect variables and their correlation patterns in multi-construct contexts.

### Previous applications with item-specific method effects

The inclusion of item-specific method effects relaxes the assumptions of the multi-state model and facilitates the identification of latent states while accounting for systematic item-specific variance in the response process. These benefits of modeling item-specific method effects have already been shown for specific constructs (i.e., cognitive components of Alzheimer's disease, children's inattention symptoms, subjective happiness, life satisfaction) in the empirical studies of Cogo-Moreira et al. [11], Geiser et al. [12], Holtmann et al. [13], and Thielemann et al. [14]. We provide an overview on the applications in Table A1 in S1 Appendix A in S1 Appendix. All applications showed that item-specific effects were present, but in the study of Geiser et al. [12] the item effects were relatively small, most likely because the indicators were constructed as rather homogenous parcels. While the applications provide valuable insights on the model implementation and the relevance of item-specific method effects in practice, they rarely investigated the item effects themselves. An exception is the analysis by Thielemann et al. [14], who included explanatory variables (i.e., gender, the highest educational degree of a person, and self-awareness) for predicting item-effect variables. However, the predictors could only partly explain variance of some item-effect variables (up to: 9% by gender, 1% by educational degree, 25% by self-awareness). Thus, item-specific method effects can be related to other person characteristics, but further investigations are warranted for their explanation.

### Different causes for item-specific method effects

When item-specific method effects are prevalent, this indicates that items are understood or dealt with differently by different persons. In the simplest case, the items differ just by a constant intercept and loading from the reference item (i.e., a congeneric model holds), then the parameters of the item-effect variables can directly be transformed to the parameters of a unidimensional model (see S1 Appendix B in S1 Appendix for the description of the special cases). Yet, when a unidimensional model does not hold, and differences between items vary between persons, then a variety of reasons can be an explanation for the item-effect variables. Items may show semantic multidimensionality, meaning that different items of a scale do not measure one unidimensional construct, but capture different aspects of a construct that are in part distinct. The idea of semantic multidimensionality is closely related to individual difference research focusing on so-called personality nuances [e.g., 22,23], where item effects can be considered secondary traits that are closely related to the focal construct but reflect unique domain content not shared with the other items. Accordingly, Holtmann et al. [13] and Thielemann et al. [14] referred to the item formulations for investigating item-specific method effects and described semantic differences.

Another explanation for item-specific method effects can be different sources of method variance, that is not conceptually related to the focal construct [see e.g., 24 for an overview], meaning that the respondents systematically differ in their use of the response scale (e.g., different response styles) or their understanding of the items (e.g., violation of measurement invariance between respondents). Such effects would be captured as item-effect variables, if

they are not constant for all items, but interact with item characteristics or content (e.g., item-length, comprehensibility, as well as positive or negative wording). For instance, Holtmann et al. [13] modeled different rater groups (i.e., self, parent, peer rating), in addition to item-specific method effects, in order to control for this method variance. Their results indicated that rater effects were not present in the application, but item-specific method effects occurred for other reasons.

A more general explanation for item-specific method effects refers to them as "systematic error" [see 25,26]. This summarizes all unique causes for an item that are stable over time, in distinction to shared causes of the items that form the states. With this explanation, different factor loadings, semantic multidimensionality, as well as method variance are included as possible causes of item-specific method effects, but other stable person characteristics may affect the responses to specific items, too. Such more general person characteristics that may interact with the item characteristics can be the familiarity of respondents with the specific item content or their motivation to answer specific items. While the term "systematic error" may lead to an interpretation that item-specific method effects are a form of error that introduces bias in relations among latent constructs [e.g., 24], they can also be referred to as "systematic item-specific variance". Especially in case of semantic multidimensionality the item-effect variables may be useful not only on psychometric grounds but for substantive analyses [see e.g., 15].

## Correlations in multi-construct multi-state models with item effects

In search of explanations for item-specific method effects, one fundamental question is, whether the effects are similar for different items. We investigate this based on the correlation among the item-effect variables and their relation to other constructs. Especially when the proposed multi-state model with item-effect variables is estimated for two (or more) different constructs, the correlations between all latent variables can be evaluated. This allows for investigations on possible explanations for item-effect variables as well as for identifying, whether item-effect variables are construct-specific or more global. Fig 1 shows an exemplary

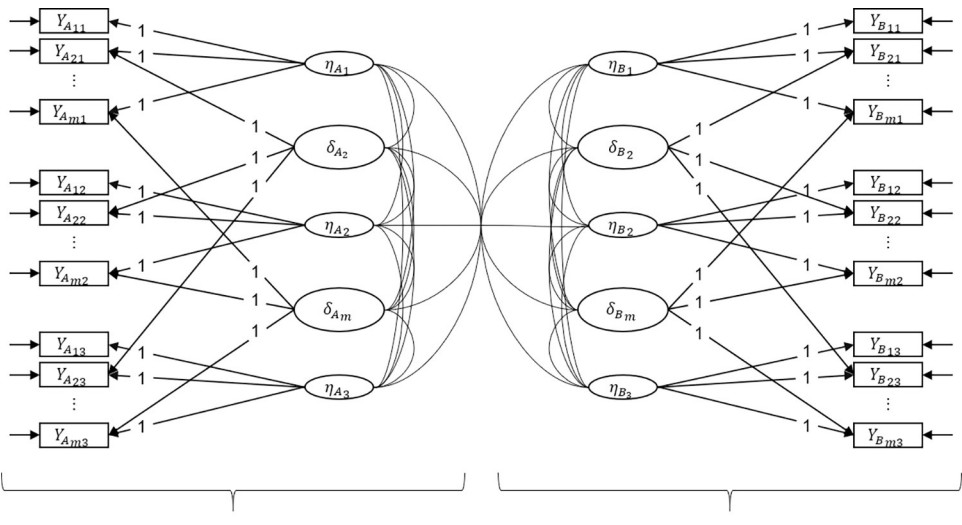

**Fig 1. A multi-state model with item-effect variables for two constructs and three measurement occasions.** Path diagram for two constructs $A$ and $B$ that are measured with $m$ manifest items $Y_{Ait}$ and $Y_{Bit}$ at three measurement occasions $t$. For each construct, three latent states $\eta_{At}$ respectively $\eta_{Bt}$ are modeled, and $m-1$ item-effect variables $\delta_{Ai}$ respectively $\delta_{Bi}$ for each item $i$, except for the reference (i.e., here the first item). Correlations between all latent variables within and between the constructs are included.

**Table 1. Exemplary correlation-matrix of an exemplary multi-state model with item-effect variables of two constructs *A* and *B*.**

|  | $\eta_{A1}$ | $\eta_{A2}$ | $\eta_{A3}$ | $\delta_{A2}$ | $\delta_{Am}$ | $\eta_{B1}$ | $\eta_{B2}$ | $\eta_{B3}$ | $\delta_{B2}$ | $\delta_{Bm}$ |
|---|---|---|---|---|---|---|---|---|---|---|
| $\eta_{A1}$ |  |  |  |  |  |  |  |  |  |  |
| $\eta_{A2}$ | (a) |  |  |  |  |  |  |  |  |  |
| $\eta_{A3}$ | (a) | (a) |  |  |  |  |  |  |  |  |
| $\delta_{A2}$ | (c) | (c) | (c) |  |  |  |  |  |  |  |
| $\delta_{Am}$ | (c) | (c) | (c) | (b) |  |  |  |  |  |  |
| $\eta_{B1}$ | (d) | (d) | (d) | (f) | (f) |  |  |  |  |  |
| $\eta_{B2}$ | (d) | (d) | (d) | (f) | (f) | (a) |  |  |  |  |
| $\eta_{B3}$ | (d) | (d) | (d) | (f) | (f) | (a) | (a) |  |  |  |
| $\delta_{B2}$ | (f) | (f) | (f) | (e) | (e) | (c) | (c) | (c) |  |  |
| $\delta_{Bm}$ | (f) | (f) | (f) | (e) | (e) | (c) | (c) | (c) | (b) |  |

*Note.* Latent states $\eta_{At}, \eta_{Bt}$ of construct A or B at different measurement occasions *t*, and item-effect variables $\delta_{Ai}, \delta_{Bi}$ of construct A or B for different items *i*. Correlations [a] within construct–among latent states, [b] within construct–among item effects, [c] within construct–latent states with item effect, [d] between constructs–among latent states, [e] between constructs–among item effects, [f] between constructs–latent states with item effects.

Typically, the relations among the states are of interest in substantive research. Correlations between the latent states *within* one construct (labeled with [a] in Table 1) are indicators of the stability of the corresponding construct across measurement occasions and correlations among the latent states *between* the constructs [d] indicate to which extent one construct can predict the other.

multi-state model for two constructs at three measurement occasions with all latent variables and their correlations. For this model, Table 1 specifies the elements of a correlation pattern.

In addition, we study the correlations of the item-effect variables. While a latent state is defined with a reference item (i.e., with no item effect), item-effect variables describe the deviation of the latent state when using another item than the reference item. Correlations among item-effect variables indicate "similarities" [14, p. 276] and can be used to identify whether items share systematic effects. When correlations with item-effect variables are zero, then no systematic item-specific variance is present or can be explained by other variables. Perfect correlations (i.e., 1 or -1) occur, when the variables perfectly explain each other. For instance, if an item differs just by a constant loading from the latent state variable, then the latent state variable itself is a perfect explanation for the occurrence of the item effect (i.e., in this case an item-effect variable is not necessary). Also, if specific items differ from the reference item due to the same reason, like only negative formulations of specific items cause differences to a positively formulated reference item, then these item-effect variables can perfectly explain each other (i.e., in this case a common method effect for the negatively formulated items would be sufficient). S1 Appendix B in S1 Appendix provides more details on the scenarios for zero and perfect correlations.

We investigate the correlation of the item-effect variables with regard to their size and systematic structure across different items and constructs. When high correlations of item-effect variables occur only *within* a construct [b], this indicates that item effects are construct-specific. For instance, they may be due to differences in factor loadings, the specific item-content, response styles, or common factors that affect the responses to the specific items. High correlations of item-effect variables *between* constructs [e] indicate that item effects can occur due to more global item and/or person characteristics, for instance, item-length, or common factors that affect the responses across constructs. Correlations of the item-effect variables with the latent states, within [c] as well as between [f] constructs, can provide first explanations for the occurrence of item effects by this construct. Hereby, high correlations of item-effect variables and latent states *within* a construct [c] indicate that the latent state is relevant for explaining

item effects and vice versa, for instance, because the items differ just in their loadings from the states. When high correlations of item-effect variables and latent states occur *between* constructs [f], this indicates that the other construct can partly explain the systematic item-specific variance.

## The present application

The described multi-construct multi-state model with item-effect variables is applied to a sample of Dutch respondents providing measures of life satisfaction and positive affect at five measurement occasions. First, we investigate the relevance of item-effect variables for both constructs considering a) model fit and b) the variance of item effects. For this we a) compare the fit of a congeneric model without item-effect variables to a model with item-effect variables and b) investigate the size and significance of the variance of each item-effect variable. Note, non-neglectable variances of the item-effect variables can occur just due to differences in factor loadings. Thus, the model comparison is essential for judging the relevance of item-effect variables. Only if constant factor loadings are not sufficient for modeling differences between the items (i.e., the congeneric model does not fit the data), it is reasonable to use the more complex model with item-effect variables that entangles different sources of systematic differences between items. Then, we use the model with item-effect variables for studying the correlation pattern (i.e., the correlations of item-effect variables with each other and with the latent state variables for life satisfaction and positive affect). This provides information about the systematic structure of item effects within and between the constructs and can indicate possible explanations for item-effect variables.

## Method

We report how we determined our sample size in a panel data analysis. The data inclusion criteria were established prior to data analysis in relation to missing values. All relevant measures, and all analyses including all tested models are made available in a public repository (see the electronic supplementary materials).

## Sample

We use data of the Dutch Longitudinal Internet Studies for the Social Sciences (LISS) panel administered by CentERdata (Tilburg University, The Netherlands), following a representative sample of the Dutch population since 2008 [27,28]. Initially 8,722 individuals were invited to participate in the panel and 6,808 did so in the first assessment wave. In total 2,596 of the participants took part in the subsequent measurements in 2009, 2011, 2013, and 2014. For our analyses, this subsample of 2,529 respondents (1,315 female) was selected, who completed at least one item on both scales of interest at each measurement occasion. In August 2008 their mean age was 49.91 years (*Min* = 16, *Max* = 88, *SD* = 14.85), and 740 (about 29%) of them had finished higher vocational education (e.g., college or university).

## Instruments

Life satisfaction was measured with the *Satisfaction with Life Scale* [29] including five items with seven-point response scales from "strongly disagree" (1) to "strongly agree" (7). The five SWLS-items were presented in Dutch as items 014–018 in the panel questionnaire (see S1 Appendix C in S1 Appendix for a detailed description of the scale). All items address the focal construct life satisfaction in a global manner. Yet, the wording and reference frame of the items differs. Item 016 assesses life satisfaction most directly and was used as the reference for

defining the latent states. As opposed, item 014 and 015 are more specific by asking for ideal and excellent life conditions. This is also the case for item 017 and 018, but these items include a retro perspective component, too. Especially item 018 seems more difficult to agree with, in comparison to the other items, as it states that one would change almost nothing in life.

Positive affect was measured with ten items of the *Positive and Negative Affect Schedule* [30] asking to what extent a person feels a certain mood in the present moment. The response scales contained seven steps from "not at all" (1) to "extremely" (7). The items were presented in Dutch as items 146–165 in the panel questionnaire (see S1 Appendix C in S1 Appendix for a detailed description of the scale). All items are formulated exactly the same way, except for the specific feeling to be rated. We used item 162 ("attentive") as the reference item for specifying positive affect in a calm and broad way. While "interested" is similar to this reference, other items are less calm, like "excited", "enthusiastic", "alert", and "active" or are more specific, like "strong", "proud", "inspired", "determined".

### Analysis strategy

First, the life satisfaction and positive affect scales were examined at each measurement occasion to evaluate whether unidimensional measurement models can be applied at each time point. Then, we used all five measurement occasions and estimated (a) a multi-state model (see Fig D.1. in S1 Appendix) and (b) a multi-state model with item-effect variables (see Fig D.2. in S1 Appendix) to investigate the relevance of item-effect variables in this application. Finally, we combined the longitudinal measurement models of the two constructs in (c) a multi-construct multi-state model with item-effect variables (see Fig D.3. in S1 Appendix) to study the correlation pattern. Path diagrams of the multi-state models are available in S1 Appendix D in S1 Appendix. All models were analyzed with the R package lavaan version 0.6–5 [31] using maximum likelihood estimation (R version 3.6.0). The R code for the analyzed models is given in the electronic supplementary materials. The fit of each model was examined in the large sample with the *Root Mean Square Error of Approximation* (RMSEA), *Standardized Root Mean Square Residual* (SRMR), and *Comparative Fit Index*, (CFI). We considered for a good/ acceptable model fit $RMSEA \leq .05/.08$, $SRMR \leq .05/.10$, $CFI \geq .97/.95$ according to Schermelleh-Engel et al. [32]. We first evaluated whether a congeneric model can be used in the data and then we investigated the variance of the item-effect variables. A neglectable variance indicates that an item-effect variable reduces to a constant that is equal across all persons (see S1 Appendix B in S1 Appendix). To identify whether specific item-effect variables are neglectable, we used the test statistic of the estimated variance divided by its standard error for testing the parameter against zero. In addition, we obtained a standardized measure to describe the size of the variance. For this, we estimated the measurement models with indicators that are standardized in their variance (i.e., each indicator was divided by its standard deviation). As such, the standard deviation of each item-effect variable can be described in standard deviation units of the respective indicator. An additive variance decomposition of the latent state and item-specific variance is not possible with this approach, as the variables can correlate.

## Results

### Single-construct models and relevance of item effects

Unidimensional models for each construct and measurement occasion reached only insufficient fits (see S1 Appendix E in S1 Appendix). Accordingly, also the multi-state models (see Fig D.1. in S1 Appendix) did not adequately describe the data, neither for life satisfaction nor for positive affect (see Table 2). In comparison to a congeneric model with unidimensional

**Table 2. Model fit parameters for the multi-state model with and without item effects for Life Satisfaction and Positive Affect across five measurement occasions.**

| Multi-state model for | $\chi^2$ | df | p-value | RMSEA [90% CI] | | SRMR | CFI |
|---|---|---|---|---|---|---|---|
| life satisfaction | | | | | | | |
| • without item effects | 7647.56 | 297 | < .001 | .10 | [.097; .101] | **.06** | .87 |
| • with item effects | 1565.73 | 271 | < .001 | **.04** | [.041; .046] | **.02** | **.98** |
| positive affect | | | | | | | |
| • without item effects | 20645.88 | 1237 | < .001 | **.08** | [.078; .080] | **.07** | .72 |
| • with item effects | 5672.10 | 1156 | < .001 | **.04** | [.038; .040] | **.03** | .94 |
| life satisfaction and positive affect | | | | | | | |
| • with item effects [+] | 9040.26 | 2574 | < .001 | **.03** | [.031; .032] | **.03** | **.95** |

*Note.* Printed in bold and italic are model fit parameters that indicate a good/acceptable model fit (*RMSEA*≤.05/.08; *SRMR*≤.05/.10; *CFI*≥.97/.95; [32]), CI = confidence interval. Variance and correlations for $\delta_{PA157}$ are fixed to 0 denoted by (+).

state variables, the presented model with correlated item-effect variables (see Fig D.2. in S1 Appendix) substantially improved model fit and resulted in an acceptable fit for both constructs with respect to the model fit indices. Thus, we examined parameter estimates of the multistate model with item-effect variables for each construct. In the following, the numbers in brackets describe the range of the estimated model parameters across measurement occasions $t$ for the state variables (i.e., $\eta_{LSt}, \eta_{PAt}$) and across items $i$ for the item-effect variables (i.e., $\delta_{LSi}$ or $\delta_{PAi}$).

**Life satisfaction (LS).** The latent states $\eta_{LSt}$ represent life satisfaction measured with the reference item 016. Their means show that on average the participants were rather satisfied with their life at all five measurement occasions ($M_{\eta_{LSt}} = [5.45; 5.58]$) but differed in their life satisfaction ($SD_{\eta_{LSt}} = [0.95; 1.03]$). The means of the item-effect variables $\delta_{LSi}$ of life satisfaction indicate that on average participants tended to reach lower scores in each other item compared to the reference item ($M_{\delta LSi} = [-0.95; -0.24]$). Moreover, the participants differed considerably in the size of these item effects ($SD_{\delta LSi} = [0.39; 0.84]$). Larger variances of item-effect variables describe larger variation among the participants in their difference to the reference item. Note, that the variances entangle various differences to the reference item (e.g., differences in factor loadings, semantic multi dimensionality with regard to the item content, method effects). All variances of the item-effect variables were significantly different from zero with $\alpha = .01$ and the standard deviations with respect to the standardized indicators were substantial ($SD^*_\delta LS_i = [0.31; 0.56]$).

**Positive affect (PA).** The latent states $\eta_{PAt}$, represent positive affect measured with the reference item 162 "attentive" and indicate that, on average, the participants tended to have a rather positive affect on all five measurement occasions ($M_{\eta_{PAt}} = [4.84; 5.08]$) with substantial inter-individual variation in their positive affect ($SD_{\eta_{PAt}} = [1.02; 1.12]$). The means of the respective item-effect variables $\delta_{PAi}$ indicate that, on average and in comparison to the reference item, participants tended to score lower on eight items (148, 150, 154, 155, 157, 159, 161, and 164) and higher on item 146 "interested" ($M_{\delta_{PAi}} = [-2.18; 0.44]$). The participants differed considerably in the size of these item effects for most of the items ($SD_{\delta PAi} = [0.47; 1.26]$); an exception is the item-effect variable $\delta_{PA157}$ of item 157 "alert", with $SD_{\delta PA157} = 0.27$. Note, that the variances entangle various differences to the reference item (e.g., differences in factor loadings, semantic multi dimensionality with regard to the item content, method effects). All variances of the latent variables were significantly different from zero with $\alpha = .01$ and the

standard deviations with respect to the standardized indicators were substantial $SD_\delta^* PA_i = [0.32; 0.86]$; except for item 157 with $SD_\delta^* PA_{157} = 0.18$. We consider less than .2 standard deviation units of a standardized indicator as a rather small standard deviation of an item-effect variable.

## Multi-construct model and correlation pattern

The multi-construct model (see Fig D.3. in S1 Appendix) allows for investigating the correlations among all latent variables within and between the two constructs life satisfaction and positive affect. We observed an acceptable fit for the multi-construct model with respect to the fit indices (see Table 2). However, the multi-construct model required a further restriction to avoid non-convergence. That is, the variance and covariances of the item-effect variable $\delta_{PA157}$, which stands out because of its small variance, were restricted to zero. All estimated means and variances for the other latent variables were nearly identical to the parameter estimates in the previously reported single-construct models, that included no restrictions on the item-effect variable $\delta_{PA157}$. All parameter estimates are summarized in Table 3.

**Correlations among latent state variables.**   The latent state variables of life satisfaction and positive affect were highly correlated within constructs across different time points $t$ and $s$ (for life satisfaction: $Cor(\eta_{LS_t}, \eta_{LS_s}) = [.54; .79]$; for positive affect: $Cor(\eta_{PA_t}, \eta_{PA_s}) = [.65; .74]$). In contrast, the latent state correlations between constructs, within and across time points were negligible to small $(Cor(\eta_{PA_t}, \eta_{LS_s}) = [.08; .25])$. Thus, both constructs, as defined with the reference item, are rather stable over the five measurement occasions, but rarely explain each other.

**Correlations among item-effect variables.**   The correlations between item-effect variables of different constructs were negligible to small across different items $i$ and $k$ (i.e., $Cor(\delta_{LS_i}, \delta_{PA_k}) = [-0.01; 0.24]$). This indicates that the item-effect variables did not generalize across the constructs and more global common causes for item-effect variables are not very plausible in this context.

Similarities of item-effect variables may be construct specific, thus, we investigated the correlations among item-effect variables within constructs. The correlations differed substantially in their size across different items within life satisfaction (i.e., $Cor(\delta_{LS_i}, \delta_{LS_k}) = [.14; .64]$) as well as within positive affect (i.e., $Cor(\delta_{PA_i}, \delta_{PA_k}) = [.16; .75]$). Substantial correlations within constructs can be related to semantic overlap of some items. Within life satisfaction $\delta_{LS014}$ and $\delta_{LS015}$, as well as $\delta_{LS017}$ and $\delta_{LS018}$ were correlated stronger, whereas the items share common contents (i.e., items 014 and 015 both refer to outstanding life and living conditions, while items 017 and 018 both refer to the life course). Within positive affect, item-effect variables, $\delta_{PA150}, \delta_{PA154}$ and $\delta_{PA155}$ had the highest correlations, whereas the items describe intense aspects of positive affect (i.e., "strong", "enthusiastic", "proud"). The lowest correlation occurred for the item-effect variables $\delta_{PA146}, \delta_{PA161}$ and the respective items are more different (i.e., "determined" and "interested").

**Correlations of latent state variables with item-effect variables.**   The correlations between the latent state variables and the item-effect variables were diverse. Between the constructs, the variables were at most weakly correlated (i.e., $Cor(\eta_{PA_t}, \delta_{LS_i}) = [-.07; .02]$ and $Cor(\eta_{LS_t}, \delta_{PA_i}) = [-.25; .25]$). Thus, the latent states defined with the item "attentive" for positive affect and "I am satisfied with my life" as a general life satisfaction measure, can hardly explain item-effect variables in another scale. Explanations within the constructs may be more informative.

Within life satisfaction the correlations were weak, too (i.e., $Cor(\eta_{LS_t}, \delta_{LS_i}) = [-.15; .14]$). This indicates that item-effect variables of the same scale, are most likely more distinct from

**Table 3. Estimated means, standard deviations and correlations of latent variables in the multi-construct multi-state model.**

| | M | SD | $\eta_{LS1}$ | $\eta_{LS2}$ | $\eta_{LS3}$ | $\eta_{LS4}$ | $\eta_{LS5}$ | $\delta_{LS014}$ | $\delta_{LS015}$ | $\delta_{LS017}$ | $\delta_{LS018}$ | $\eta_{PA1}$ | $\eta_{PA2}$ | $\eta_{PA3}$ | $\eta_{PA4}$ | $\eta_{PA5}$ | $\delta_{PA146}$ | $\delta_{PA148}$ | $\delta_{PA150}$ | $\delta_{PA154}$ | $\delta_{PA155}$ | $\delta_{PA157}$ | $\delta_{PA159}$ | $\delta_{PA161}$ |
|---|---|---|---|---|---|---|---|---|---|---|---|---|---|---|---|---|---|---|---|---|---|---|---|---|
| $\eta_{LS1}$ | 5.58 | 0.95 | | | | | | | | | | | | | | | | | | | | | | |
| $\eta_{LS2}$ | 5.54 | 0.95 | .74 | | | | | | | | | | | | | | | | | | | | | |
| $\eta_{LS3}$ | 5.53 | 0.99 | .67 | .72 | | | | | | | | | | | | | | | | | | | | |
| $\eta_{LS4}$ | 5.50 | 1.02 | .59 | .66 | .73 | | | | | | | | | | | | | | | | | | | |
| $\eta_{LS5}$ | 5.45 | 1.03 | .54 | .63 | .70 | .79 | | | | | | | | | | | | | | | | | | |
| $\delta_{LS014}$ | -0.46 | 0.41 | .04 | .07 | .06 | .06 | .06 | | | | | | | | | | | | | | | | | |
| $\delta_{LS015}$ | -0.32 | 0.39 | .07 | .07 | .09 | .09 | .14 | .64 | | | | | | | | | | | | | | | | |
| $\delta_{LS017}$ | -0.24 | 0.51 | -.09 | -.12 | -.13 | -.15 | -.15 | .28 | .18 | | | | | | | | | | | | | | | |
| $\delta_{LS018}$ | -0.95 | 0.84 | .05 | .06 | .02 | .03 | .03 | .31 | .14 | .43 | | | | | | | | | | | | | | |
| $\eta_{PA1}$ | 5.08 | 1.03 | .19 | .12 | .14 | .13 | .11 | .02 | -.06 | -.04 | -.01 | | | | | | | | | | | | | |
| $\eta_{PA2}$ | 4.96 | 1.07 | .10 | .14 | .12 | .08 | .12 | -.01 | -.07 | -.04 | .00 | .73 | | | | | | | | | | | | |
| $\eta_{PA3}$ | 4.85 | 1.07 | .11 | .11 | .20 | .14 | .14 | -.03 | -.06 | -.06 | -.03 | .68 | .72 | | | | | | | | | | | |
| $\eta_{PA4}$ | 4.84 | 1.12 | .12 | .12 | .18 | .24 | .20 | .01 | -.04 | -.03 | -.02 | .67 | .67 | .73 | | | | | | | | | | |
| $\eta_{PA5}$ | 4.90 | 1.09 | .10 | .10 | .16 | .18 | .25 | .01 | -.06 | -.07 | -.04 | .65 | .66 | .73 | .74 | | | | | | | | | |
| $\delta_{PA146}$ | 0.44 | 0.76 | .07 | .10 | .06 | .08 | .07 | .08 | .01 | .04 | .03 | -.47 | -.49 | -.48 | -.45 | -.45 | | | | | | | | |
| $\delta_{PA148}$ | -2.18 | 1.26 | -.18 | -.19 | -.25 | -.23 | -.24 | .18 | .09 | .10 | .09 | -.53 | -.57 | -.59 | -.58 | -.56 | .36 | | | | | | | |
| $\delta_{PA150}$ | -0.13 | 0.81 | .21 | .25 | .18 | .19 | .18 | .19 | .16 | .03 | .08 | -.37 | -.41 | -.43 | -.41 | -.42 | .40 | .39 | | | | | | |
| $\delta_{PA154}$ | -0.38 | 0.82 | .21 | .24 | .17 | .19 | .20 | .24 | .09 | .00 | .09 | -.32 | -.34 | -.33 | -.32 | -.30 | .56 | .49 | .65 | | | | | |
| $\delta_{PA155}$ | -0.36 | 1.03 | .16 | .19 | .14 | .15 | .13 | .24 | .06 | .08 | .13 | -.29 | -.34 | -.33 | -.33 | -.31 | .34 | .50 | .70 | .75 | | | | |
| $\delta_{PA157}$ | -0.25 | (+) | (+) | (+) | (+) | (+) | (+) | (+) | (+) | (+) | (+) | (+) | (+) | (+) | (+) | (+) | (+) | (+) | (+) | (+) | (+) | | | |
| $\delta_{PA159}$ | -0.99 | 0.80 | .07 | .09 | .05 | .05 | .07 | .16 | .06 | .01 | .13 | -.18 | -.21 | -.19 | -.22 | -.18 | .34 | .42 | .39 | .64 | .45 | (+) | | |
| $\delta_{PA161}$ | -0.30 | 0.55 | .14 | .17 | .11 | .14 | .12 | .16 | .11 | .05 | .13 | -.07 | -.12 | -.15 | -.12 | -.13 | .16 | .23 | .48 | .49 | .49 | (+) | .46 | |
| $\delta_{PA164}$ | -0.22 | 0.67 | .22 | .23 | .18 | .21 | .19 | .15 | .11 | -.01 | .07 | -.20 | -.22 | -.22 | -.20 | -.21 | .42 | .25 | .46 | .58 | .37 | (+) | .53 | .40 |

*Note.* Shown are the mean (*M*), standard deviation (*SD*) and correlation-matrix for the latent states of the Satisfaction with Life Scale $\eta_{LSi}$, and the Positive Affect Schedule $\eta_{PAt}$ at different measurement occasions *t*, and the latent item effects $\delta_{LSi}$, $\delta_{PAt}$ of the constructs for different items *i*. Printed in italic are the correlations that are *not* significantly different from 0 with $\alpha = .01$ in the large sample. SD and correlations for $\delta_{PA157}$ are fixed to 0 denoted by (+).

the general measure of life-satisfaction. Instead, within positive affect, the latent state variables were moderately to highly negative correlated with some item-effect variables (i.e., $Cor(\eta_{PA_t}, \delta_{PA_i}) = [-.59; -.29]$ for $\delta_{PA146}, \delta_{PA148}, \delta_{PA150}, \delta_{PA154}$, and $\delta_{PA155}$), and weakly correlated with others (i.e., $Cor(\eta_{PA_t}, \delta_{PA_i}) = [-.22; -.07]$ for $\delta_{PA159}, \delta_{PA161}$, and $\delta_{PA164}$). Thus, 'attentive' positive affect can partly explain some item-effect variables and vice versa. For these item-effect variables, differences in factor loadings can be a plausible explanation, as well as common causes of the item-effect variables and 'attentive' affect.

**Correlation pattern.** For the specified item-effect variables in the SWLS, we obtained no substantial correlation to the variables of the positive affect scale (i.e., states or item-effect variables between constructs) or to the general measure of life satisfaction (i.e., states within the construct). Only for some item-effect variables within the construct, we found similarities in relation to the specific item content. These result support the interpretation of item-effect variables in the sense of semantic multidimensionality (i.e., the item-effect variables reflect unique domain content for the focal construct in relation to the item content). Yet, other explanations cannot be ruled out, like the impact of more general person characteristics or method effects that occur only in relation to the specific item content. The correlation pattern is a bit different for the specified item-effect variables in the positive affect scale. Like for life satisfaction, all substantial correlations were rather construct-specific and more global common causes of item-effect variables are less likely (i.e., no substantial correlations to states or item-effect variables between constructs). However, for some items, the item-effect variables were substantially correlated to the states within the construct. In addition, we found some similarities among the item-effect variables within the construct in relation to the item content. As such, semantic multidimensionality may be present for some items, but possibly also differences in relation to the states (i.e., different factor loadings, or more substantive explanations for item-effect variables due to 'attentive' positive affect as well as related constructs). Again, further explanations for item-effect variables cannot be ruled out.

## Discussion

Perfectly unidimensional measurements might be an unobtainable goal in many applied settings. Rather, specific items can exhibit additional systematic variations reflecting item-specific individual differences beyond the measured common state. The systematic item-specific variance can be captured as item-effect variables. We introduced a multi-construct multi-state model for longitudinal data that allows for modeling item-effect variables and for studying their correlation pattern. When item-effect variables correlate across different constructs, this can indicate more global explanations, like systematic response styles. While construct specific correlations of item-effect variables possibly refer to different factor loadings or semantic multidimensionality.

### Empirical insights

In our application using measures of life satisfaction and positive affect across five measurement occasions, we demonstrated the importance of acknowledging item-effect variables. A congeneric model could not describe the item responses, but the inclusion of item-effect variables substantially improved model fit. In addition, all but one of the item-effect variables showed a non-negligible variance. Thus, differences between items were not constant across the respondents, but modeling individual differences could describe the response process.

Our attempt in identifying systematic associations between the item-effect variables within and between constructs was less successful in the specific setting, because the correlation pattern was complex and various explanations for item-effect variables are possible. All

correlations between the constructs were at most weak, indicating only slight similarities between the constructs and no common causes for the observed item-effect variables. Within the constructs, the correlation pattern differed. For life satisfaction, few item-effect variables exhibited high correlations among each other, but not with the latent states. Where high correlations occurred, they were plausible in relation to the item content and depict rather distinct aspects from general life satisfaction measured with the states. A possible explanation for this pattern can be that the item-effect variables reflect unique domain content (i.e., semantic multidimensionality), like in individual difference research focusing on so-called personality nuances [e.g., 22,23]. Within positive affect, the latent states were specified as 'attentive', this construct partly explained the item-effect variables. The substantial correlation can indicate differences in factor loading between items, but the pattern is not clear and a congeneric model did not hold in the application. Given that a state measure of positive affect was administered, one might also speculate that some item-effect variables are a consequence of individual differences in attention (or lack thereof) when responding to the items. Yet, this association did not generalize to all items of the scale and, importantly, did not transfer to the second construct. Thus, the observed item-effect variables seemed to have different causes and were rather item-specific. This might be different for other instruments with different item content. For instance, when item-effect variables generalize more across constructs, this may be caused by common method variance like a negative wording effect [e.g., 24]. As such, other applications might exhibit a clearer correlation pattern of item-effect variables. Such a pattern could be explained by other explanatory variables, such as psychological constructs (e.g., self-awareness like in the study of Thielemann et al. [14]), but also group membership, test behavior (e.g., response style), or attitudes towards psychological tests.

## Strengths and limitations

The presented multi-state multi-construct model with item-effect variables defines all latent variables based on LST theory. Item-effect variables are modelled as stable differences between an item and the reference item, and we only use necessary model assumptions that specify measurement invariance across time points. Thus, all latent correlations can be further investigated. However, in this flexible model, various explanations for item-effect variables are possible. We provide first insights for investigating similarities and differences between item-effect variables within and between constructs based on the correlation pattern. This can help to identify whether common causes for the systematic item-specific variation are plausible. Especially if this is the case for different constructs, the nature of item-effect variables is more global and less related to the measurement of a specific construct. This was not the case in our application, but it illustrates the interesting information provided in the correlation pattern.

Based on this, item-effect variables in different scales can be further investigated in subsequent research. For instance, in psychometric research on scale construction, our model can be used for studying the dimensionality in multi-construct scales and for identifying, whether more strict measurement assumptions are plausible (e.g., for modelling a common method effect for specific items). Also, methodological research can evaluate the occurrence of item-effect variables under different conditions (e.g., in relation to the item format, different response scales or item characteristics) and can investigate whether item effects generalize across different constructs. Especially substantive research (e.g., on the contribution of item-effect variables for predictive validity) would gain from a better understanding on similarities and possible explanations of item-effect variables [e.g., 15]. As this can help for judging whether the item-effect variables represent semantic or method variance in relation to the item content, and respectively may be more or less useful for substantive analysis.

For investigations on the item-effect variables, it is very important to keep in mind that we used a specific modelling approach relying on a reference coding scheme. Thus, the interpretation of all latent variables refers to this specification and can be different when using another identification scheme or modelling approach with different assumptions.

## Alternative modeling approaches

For using the reference coding scheme, the selection of the reference item should be plausible for a specific instrument; we relied on substantive theory and considered items that captured the corresponding construct most directly, although this was less clear for the positive affect scale. Using another reference item will not affect model fit, but the interpretation of the latent variables. Similar, an alternative parametrization using an effect coding scheme can be applied for specifying average states across all items [see 14,33]. The item-effect variables will always represent the difference to the latent states, but the interpretation of the item-effect variables depends on the meaning of the latent states. Thus, analyzing the same data with another identification scheme will enable different interpretations in subsequent analyses. Our approach of using a global refence item might support the occurrence of item-effect variables that capture more distinct aspects of other items in the sense of semantic multi-dimensionality. Instead using a more specific reference item may results in more similarities between the item-effect variables (i.e., they share variation from a specific item). The specification of common latent states would entangle content of different items, thus, comparisons between specific items would be limited.

The presented approach poses no restrictions on the correlation structure of item-effect variables and states, but entangles differences in factor loadings with multidimensionality between items. Modeling item effects as residuals typically can include different factor loadings and, thus, can help to disentangle the variance components [see also e.g., 7,12]. However, typically the correlations of item effects and states of the same construct are restricted to zero in these models for additive variance decomposition (i.e., for investigations on the consistency and method-specificity of items). This assumption can be violated in practice. Geiser and Lockhart [4] provide a detailed treatment of similarities and differences between the latent residual and our difference variable approach for defining item-specific method effects. Another modeling approach would be the specification of item-specific traits [e.g., 7–10] that disentangles the situation specificity from the measures. This approach would directly depict a multidimensional construct specification with different items. However, it does not allow for investigations on the item-specific effects itself, because common and item-specific components are entangled. Depending on the causes of item-specific method effects, different modeling approaches can be reasonable, and our approach can help to gain some insights on possible explanations

When choosing a modeling approach, the scale level of the indicators must be considered. In our application, we relied on the assumption that the items can be treated as continuous, that is the rating scales are rated with sufficient gradations [e.g., 34]. For dichotomous data, the same modeling approach in the tradition of item-response theory can be considered [see 14]. Strategies for modeling item-effect variables with ordinal data are currently under investigation and require additional assumptions on the invariance of threshold parameters across measurement occasions. Yet, already Holtmann et al. [13] presented an approach of modeling item effects for ordinal responses as residuals.

## Conclusion

The present study presented an approach for modeling item-effect variables in longitudinal data using multi-construct multi-state models. Model fit and the variances of the item-effect

variables provide the basis to check whether item-specific interindividual differences are present when measuring a common latent variable with different items. In addition, the correlation pattern within and between constructs can be examined for systematically studying item-effect variables in different scales and settings. For this we provide first empirical insights, but further applications would be beneficial.

## Supporting information

**S1 Appendix.**

- Appendix A: Overview on recent applications with item-specific method effects

- Appendix B: Different scenarios for zero or perfect correlations with item-effect variables

- Appendix C: Description of the scales

- Appendix D: Path diagrams

  ○ Path diagrams of the multi-state models of Life Satisfaction

  ○ Path diagram of the common model of Life Satisfaction and Positive Affect

- Appendix E: Model fit for unidimensional models.
  (DOCX)

**S1 File.**
(DOCX)

## Author Contributions

**Conceptualization:** Tina H. Erhardt, Timo Gnambs, Marie-Ann Sengewald.

**Formal analysis:** Marie-Ann Sengewald.

**Methodology:** Tina H. Erhardt, Marie-Ann Sengewald.

**Supervision:** Timo Gnambs, Marie-Ann Sengewald.

**Visualization:** Tina H. Erhardt.

**Writing – original draft:** Tina H. Erhardt.

**Writing – review & editing:** Timo Gnambs, Marie-Ann Sengewald.

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
