## [Decision Letter · Decision Letter 0]

2 Sep 2022

PONE-D-22-10802Studying item-effects and their correlation patterns with multi-construct multi-state modelsPLOS ONE

Dear Dr. Sengewald,

Thank you for submitting your manuscript to PLOS ONE. After careful consideration, we feel that it has merit but does not fully meet PLOS ONE’s publication criteria as it currently stands. Therefore, we invite you to submit a revised version of the manuscript that addresses the points raised during the review process.

Please see the comments from two reviewers below and in two attachments. Both reviewers are positive about the manuscript, and have provided specific feedback that may improve the presentation of the study.

We look forward to receiving your revised manuscript.

Kind regards,

Hanna Landenmark

Staff Editor

PLOS ONE

Journal Requirements:

Reviewers' comments:

Reviewer's Responses to Questions

**Comments to the Author**

1. Is the manuscript technically sound, and do the data support the conclusions?

Reviewer #1: Yes

Reviewer #2: Yes

2. Has the statistical analysis been performed appropriately and rigorously? 

Reviewer #1: Yes

Reviewer #2: Yes

3. Have the authors made all data underlying the findings in their manuscript fully available?

Reviewer #1: Yes

Reviewer #2: No

4. Is the manuscript presented in an intelligible fashion and written in standard English?

Reviewer #1: Yes

Reviewer #2: No

5. Review Comments to the Author

Reviewer #1: The manuscript presents a multi-construct multi-state model that can be estimated from data on sets of items that measure multiple constructs that are administered at mutliple timepoints. In this way, item-effect variables can be estimated which are defined in line with Pohl (2008) as the difference between a latent-state variable that is specific to an item, and the common latent-state variable, which is the latent-state variable of a chosen reference item. The authors then apply this model to a dataset with items that measure positive affect and items that measure life satisfaction. I like that this model can help study the factors that contribute to systematic variance that is not captured in the common latent state. In general, I think this manuscript can be a nice addition to the literature on item effects. Below, I include several points and questions of clarification to help improve the presentation of the content.

I have uploaded my comments as a pdf attachment

Reviewer #2: The primary findings and approach in the manuscript are sound.

The statistical analyses seem to have been performed correctly.

The data underlying findings is not provided, though the authors describe the reason for this in their materials. Although the manuscript is well-written in most sections, there are some parts of it that need further attention before publication. See the attached document for further review comments.

6. PLOS authors have the option to publish the peer review history of their article (what does this mean?). If published, this will include your full peer review and any attached files.

Reviewer #1: No

Reviewer #2: **Yes: **Kaylee Litson

---

## [Author Response · Author response to Decision Letter 0]

9 Jan 2023

Thank you very much for the positive feedback on the merit of our manuscript and the valuable suggestions for the revision. We carefully went through the manuscript and corrected all points raised by the reviewers. We provide detailed responses to all comments of the editor and the two reviewers in the document "Response to Reviewers".

---

## [Decision Letter · Decision Letter 1]

17 May 2023

PONE-D-22-10802R1Studying item-effect variables and their correlation patterns with multi-construct multi-state modelsPLOS ONE

Dear Dr. Sengewald,

Thank you for submitting your manuscript to PLOS ONE. After careful consideration, we feel that it has merit but does not fully meet PLOS ONE’s publication criteria as it currently stands. Therefore, we invite you to submit a revised version of the manuscript that addresses the minor points raised during the review process. 

I was asked to guest edit this manuscript after my initial review as Reviewer 2. The paper has addressed both mine and Reviewer 1’s comments appropriately, and I enjoyed reading the revised manuscript. Like Reviewer 1, I appreciate the new addition of Appendix B and its overview of the modeling approach. Further, the expanded, edited introduction and discussion sections highlight the relevance of this new approach, and the additional clarity in results makes a strong contribution to this piece. Only minor revisions are requested prior to publication, and the necessary revisions are noted in the comments below.

Reviewer 1 describes an important concern about the mathematical configuration of the single construct models without item-specific factors that are presented in Appendix B (please see their review for an entire overview). They note, and I also see, that the variance of delta is intertwined with differences in factor loadings within the single construct models without item specific effects. They then make a suggestion about regressing the items on the reference item and changing delta to be the residual. This could be an important avenue for future work, yet in my own reading of this paper, this factor loading issue they raise is only relevant in the single construct models without item specific effects; in the manuscript, you seem to interpret the single construct models with item specific effects, as well as the multi-construct models with item specific effects, which have all factor loadings constrained to 1. Thus, this issue about factor loadings impacting correlation coefficients does not need to be addressed in the revision of the current manuscript, unless you feel inclined to address it.

That said, it was difficult to determine which exact model was being interpreted in the results, since there is a lot of specific terminology surrounding these modeling approaches. The paper would benefit from referencing the exact Appendix Figures in the results section when discussing each set of results. Please reference these figures in your revised manuscript.

In addition to this minor edit, there are a few minor line items Reviewer 1 noted. I also have a few minor line comments. All minor comments should be addressed in the final revision.

The OSF link did not load, so please check that it is made accessible upon publication.Line 449-450 “Yet, other explanations cannot be ruled out.” Can you elaborate on this a little bit more. What competing explanations cannot be ruled out?Line 510 has an extra or missing word. Please revise.

We look forward to receiving your revised manuscript.

Kind regards,

Kaylee Litson, Ph.D.

Guest Editor

PLOS ONE

Journal Requirements:

Reviewers' comments:

Reviewer's Responses to Questions

**Comments to the Author**

1. If the authors have adequately addressed your comments raised in a previous round of review and you feel that this manuscript is now acceptable for publication, you may indicate that here to bypass the “Comments to the Author” section, enter your conflict of interest statement in the “Confidential to Editor” section, and submit your "Accept" recommendation.

Reviewer #1: (No Response)

2. Is the manuscript technically sound, and do the data support the conclusions?

Reviewer #1: Yes

3. Has the statistical analysis been performed appropriately and rigorously? 

Reviewer #1: Yes

4. Have the authors made all data underlying the findings in their manuscript fully available?

Reviewer #1: Yes

5. Is the manuscript presented in an intelligible fashion and written in standard English?

Reviewer #1: Yes

6. Review Comments to the Author

Reviewer #1: I enjoyed reading this revised manuscript which reads very well now. I have one more comment and some very minor things that I attach as a pdf.

7. PLOS authors have the option to publish the peer review history of their article (what does this mean?). If published, this will include your full peer review and any attached files.

Reviewer #1: No

---

## [Author Response · Author response to Decision Letter 1]

8 Jun 2023

We greatly appreciate the careful second review round. In the revision, we carefully considered all points and provide detailed responses to all comments of the guest editor and reviewer in the file Response to Reviewers. We hope you share our impression that the present version meets PLOS ONE’s publication criteria.

---

## [Editor Report · Decision Letter 2]

4 Jul 2023

Studying item-effect variables and their correlation patterns with multi-construct multi-state models

PONE-D-22-10802R2

Dear Dr. Sengewald,

We’re pleased to inform you that your manuscript has been judged scientifically suitable for publication and will be formally accepted for publication once it meets all outstanding technical requirements.

Kind regards,

Kaylee Litson, Ph.D.

Guest Editor

PLOS ONE

Additional Editor Comments (optional):

Thank you very much for your interesting paper. Please do a final proof read of the paper and address any grammatical or spelling errors, as well as ensure all citations are included as appropriate. Congratulations, and I look forward to seeing this paper in press!

---

## [Editor Report · Acceptance letter]

11 Aug 2023

PONE-D-22-10802R2 

Studying item-effect variables and their correlation patterns with multi-construct multi-state models 

Dear Dr. Sengewald:

I'm pleased to inform you that your manuscript has been deemed suitable for publication in PLOS ONE. Congratulations! Your manuscript is now with our production department. 

Kind regards, 

on behalf of

Dr. Kaylee Litson 

Guest Editor

PLOS ONE